# Prediction of Post-Bath Body Temperature Using Fuzzy Inference Systems with Hydrotherapy Data

**DOI:** 10.3390/healthcare13090972

**Published:** 2025-04-23

**Authors:** Feng Han, Minghui Tang, Ziheng Zhang, Kenji Hirata, Yoji Okugawa, Yunosuke Matsuda, Jun Nakaya, Katsuhiko Ogasawara, Kohsuke Kudo

**Affiliations:** 1Department of Diagnostic Imaging, Graduate School of Medicine, Hokkaido University, Kita-Ku, Sapporo 060-8638, Hokkaido, Japan; 2Department of Diagnostic Imaging, Faculty of Medicine, Hokkaido University, Kita-Ku, Sapporo 060-8638, Hokkaido, Japan; 3Medical AI Research and Development Center, Hokkaido University Hospital, Kita-Ku, Sapporo 060-8648, Hokkaido, Japan; 4Product Development Department, Bathclin Corporation, Higashiarai, Tsukuba 305-0033, Ibaraki, Japan; 5Tenshi Hospital, Higashi-Ku, Sapporo 065-0012, Hokkaido, Japan; 6Department of Health Sciences and Technology, Faculty of Health Sciences, Hokkaido University, N12-W5, Kita-Ku, Sapporo 060-0812, Hokkaido, Japan

**Keywords:** hydrotherapy, body temperature prediction, artificial intelligence, fuzzy inference systems, machine learning, non-invasive monitoring, physiological prediction

## Abstract

**Background/Objectives:** Widely known for its therapeutic benefits, hydrotherapy utilizes water’s physical properties, such as temperature, hydrostatic pressure, and viscosity, to influence physiological responses. Among these, body temperature modulation plays a crucial role in enhancing circulatory function, muscle relaxation, and metabolic processes. While hydrotherapy can improve systemic health, particularly cardiac function, improper temperature control poses risks, especially for vulnerable populations such as the elderly or individuals with thermoregulatory impairments. Therefore, accurately predicting post-bath body temperature is essential for ensuring safety and optimizing therapeutic outcomes. **Methods:** This study explored the use of fuzzy inference systems to predict post-bath body temperature, leveraging an adaptive neuro-fuzzy inference system, evolutionary fuzzy inference system (EVOFIS), and enhanced Takagi-Sugeno fuzzy system. These models were compared with random forest and support vector machine models using hydrotherapy-related data. **Results:** The results show that EVOFIS outperformed other models, particularly in predicting deep body temperature, which is clinically significant as it is closely linked to core physiological regulation. **Conclusions:** The ability to accurately forecast deep-temperature dynamics enables proactive management of hyperthermia risk, supporting safer hydrotherapy practices for at-risk groups. These findings highlight the potential of FIS-based models for non-invasive temperature prediction, contributing to enhanced safety and personalization in hydrotherapy applications.

## 1. Introduction

Hydrotherapy is a therapeutic intervention that capitalizes on water’s intrinsic physical properties—such as temperature, hydrostatic pressure, and viscosity—to achieve diverse therapeutic outcomes [1]. Among these mechanisms, body temperature modulation plays a crucial role in regulating physiological processes [2]. Body temperature, typically categorized into surface temperature and deep body temperature, contributes uniquely to hydrotherapy’s therapeutic efficacy. Surface temperature, influenced by external features such as water and air temperatures, directly impacts thermoregulation and blood circulation. For example, increased surface temperature induces vasodilation, enhancing blood flow and facilitating heat dissipation [3]. These physiological changes are instrumental in reducing muscle stiffness and improving joint mobility, underscoring their therapeutic value [4]. In contrast, deep body temperature reflects the thermal state of internal organs and central systems and is tightly regulated to maintain homeostasis. Changes in deep body temperature can significantly influence blood flow, cardiovascular responses, and autonomic nervous system activity. For instance, elevated core temperature promotes peripheral vasodilation, reduces vascular resistance, and modulates sympathetic tone—effects that are particularly relevant for individuals with circulatory or thermoregulatory challenges. Hydrotherapy’s ability to influence deep body temperature has been linked to systemic health benefits, such as improved cardiac function in patients with congestive heart failure [5]. However, in certain vulnerable populations, including individuals with pre-existing health conditions or impaired thermoregulation, even slight deviations in deep body temperature can lead to significant physiological consequences, such as cognitive impairment, fatigue, or heat-related illnesses, such as heat stroke [6]. Therefore, accurately monitoring and managing deep body temperature are crucial for optimizing hydrotherapy interventions [7].

In clinical practice and research, surface temperature is typically measured via thermometric devices, while deep body temperature often requires invasive or specialized equipment. To overcome these practical challenges, predictive models are frequently employed. For instance, in one study, the authors utilized infrared thermography to measure facial surface temperatures and applied regression algorithms, achieving a prediction error of ±0.285 °C and a mean absolute error of 0.240 °C [8]. Similarly, deep body temperature has been predicted using non-contact skin temperature and heat flux data with an error between 0.28 °C and 0.34 °C under varying environmental conditions [9]. Additionally, a dual-temperature sensor system measuring ear canal surface temperature was developed, achieving a prediction error of ±0.49 °C for deep body temperature in diverse scenarios [10]. Despite these advancements, previous studies have primarily focused on the real-time estimation of current temperature states, often neglecting the prediction of future temperature. This is particularly problematic in hydrotherapy, where thermal responses continue to evolve after immersion, and post-bath temperature dynamics play a key role in patient safety—especially in elderly populations [7]. While these methods provide useful insights for instantaneous assessment, they fall short in predictive applications. Techniques such as infrared thermography primarily measure surface-level temperature at a single time point and do not account for the temporal dynamics of body temperature regulation. Moreover, they are limited in their ability to model internal thermal responses, such as those involving deep body temperature changes or delayed heat transfer processes. These limitations underscore the need for models that can integrate time-dependent physiological signals and external factors—such as bathing duration, water temperature, and pre-existing thermal states—to provide reliable predictions over clinically relevant timeframes.

Recent advances in artificial intelligence (AI) have further expanded the potential for predictive healthcare applications, including the non-invasive monitoring and dynamic modeling of physiological states [11]. For example, AI-based frameworks have demonstrated success in predicting patient outcomes, identifying clinical risks, and optimizing treatment plans [12]. The integration of AI into healthcare presents promising opportunities for addressing long-standing challenges in clinical practice. Among AI-driven methodologies, knowledge-driven approaches such as fuzzy inference systems (FISs) offer a promising solution for the complex task of predicting deep body temperature in hydrotherapy. Unlike data-driven machine learning techniques such as random forest (RF) and support vector machine (SVM), FISs integrate rule-based logic with expert knowledge, enhancing their interpretability and robustness [13]. By accommodating uncertainty and modeling complex, nonlinear interactions, FISs effectively capture the intricate dynamics of temperature regulation influenced by multiple features, such as age, environmental conditions, and water composition. Building upon the foundational advantages of FISs, advanced variants such as the adaptive neuro-fuzzy inference system (ANFIS), evolutionary fuzzy inference system (EVOFIS), and enhanced Takagi-Sugeno fuzzy system (eTS) have further enhanced performance across domains including meteorology and industrial automation [14,15]. Furthermore, FIS-based methods have consistently outperformed neural networks in both prediction accuracy and computational efficiency [16,17], reinforcing their viability as robust alternatives for predictive modeling in healthcare applications, particularly in physiological state estimation. On the basis of prior evidence and their interpretability, FIS methods were selected for their ability to model nonlinear physiological processes while offering transparent, clinically relevant outputs.

In this study, we employed FIS methodologies to predict post-bath body temperature (both surface and deep) and evaluated the predictive performance of various FIS models compared with traditional machine learning techniques. Additionally, key features influencing temperature dynamics were analyzed. The insights gained from this research are expected to enhance the safety and efficacy of hydrotherapy interventions.

## 2. Materials and Methods

This study followed a structured methodology comprising three main steps (Figure 1): data preprocessing, feature selection, and predictive modeling. First, hydrotherapy data were collected and preprocessed, including the removal of missing data and other necessary data-cleaning steps to ensure dataset integrity. Next, six feature selection methods were applied to identify the most relevant features influencing body temperature changes. Finally, three fuzzy inference system-based models (ANFIS, EVOFIS, and eTS) and two traditional machine learning models (RF and SVM) were used for prediction, and their performance was evaluated to determine the most effective approach. This study was conducted in accordance with the Declaration of Helsinki and was approved by the Institutional Review Board of Hokkaido University Hospital (protocol code #024-0475, approved on 12 March 2025). The overall machine learning system architecture is illustrated in Figure 1, which presents the entire workflow from data preprocessing to prediction.

### 2.1. Dataset

The dataset, provided by Bathclin Corporation (Tokyo, Japan), includes hydrotherapy-related records collected from 1991 to 1997 and comprises 213 rows (records) and 25 columns (variables) (Figure 2). The 25 variables are gender, bathing time, bath temperature, and pre- and post-bath temperatures recorded at 10, 20, 30, and 60 min for 4 body parts: right-hand skin temperature, left-hand deep temperature, forehead skin temperature, and left-hand non-contact surface temperature (Appendix A). The 60 min post-bath temperature was selected as the target variable, while the remaining variables served as input features. This dataset was well structured and rich in temperature data across multiple body sites and time intervals, making it well suited for modeling temporal physiological changes.

### 2.2. Feature Selection

To minimize noise, reduce redundancy, and enhance computational efficiency [18], we first conducted a preprocessing step prior to formal feature selection. Variables with excessive missing values (e.g., temperature at 0 min), those with constant values across all samples, or those clearly irrelevant to the target variable (e.g., subject ID or recording date) were excluded. In addition, records with missing key variables or implausible temperature values were removed. Specifically, body temperatures exceeding 38 °C—a common clinical threshold for fever [19]—were excluded to avoid the influence of potential pathological conditions. The original dataset contained 244 records, and after preprocessing, 213 valid records were retained for analysis. As a result, a refined set of 25 candidate features was used in subsequent modeling.

Subsequently, six key predictive features were selected using multiple feature selection techniques, including Lasso regularization (L1 Norm) [20], Ridge regularization (L2 Norm) [21], random forest (RF) [22], and hybrid approaches combining these methods with recursive feature elimination (RFE) [23], denoted as L1 + RFE, L2 + RFE, and RF + RFE (Figure 1). The specific features selected by each method are detailed in the Results section.

### 2.3. Model Training

Three fuzzy logic-based models—ANFIS, eTS, and EVOFIS—were employed, and their performance was compared with that of RF and SVM.

For ANFIS, each feature was represented using two Gaussian membership functions with the trainable parameters µ and *σ*, initialized as [0.0, 1.0] and [1.0, 1.0], respectively. Optimization was performed using the Adam algorithm with a learning rate of 0.005, and mean absolute error (L1 Loss) was used as the loss function. The model was trained for 1000 epochs. For eTS, the learning rate was set to 0.1, and initial weights were fixed at 1000. The model dynamically adjusted its parameters to data variations during training. For EVOFIS, the Optuna (PrefferdNetworks Inc., Tokyo, Japan, version 3.6.1) framework was utilized to optimize the following hyperparameters for the differential evolution algorithm, with search spaces: mutation rate (0.0–1.0), crossover probability (0.0–1.0), population size range (30–60), range of generations (1500–2000).

The RF model was constructed using 100 decision trees and a random seed of 42 to ensure reproducibility. For the support vector regression (SVR) model, the regularization parameter (C), kernel coefficient (*γ*), and epsilon-tube (ε) were optimized using GridSearchCV from Scikit-learn (version 1.3.0). The parameter grid included: *C*: [0.1, 1, 10, 100]; γ: [1 × 10^−4^, 1 × 10^−3^, 1 × 10^−2^, 1 × 10^−1^, ’scale’]; and ε: [0.1, 0.2, 0.5, 1]. To prevent overfitting and improve generalization, 5-fold cross-validation was applied during hyperparameter tuning, evaluating the negative mean squared error on the training set. The optimal hyperparameter combination was then used to train the final SVR model, which was subsequently applied to the test set for predictions.

A total of 180 records after feature selection were randomly selected for training, while the remaining 33 records were reserved for testing. Additionally, Gaussian membership functions and convergence curves were obtained to interpret model training dynamics. All algorithms were implemented in a Windows 11 (version 23H2) environment using Python (version 3.11.4) on a system with the following configuration: a 12th Gen Intel(R) Core (TM) i7-12700K processor running at 3.60 GHz, with 16 GB of RAM, and an NVIDIA GeForce RTX 3060Ti GPU. All experiments were conducted using publicly available libraries and explicitly defined parameter settings to ensure reproducibility.

### 2.4. Evaluation

Two metrics were used to evaluate model performance: the coefficient of determination (R^2^) and the mean squared error (*MSE*). The MSE formula is as follows:(1)$$MSE=\frac{1}{n}*\sum (yi-ŷi)²$$
where n is the number of samples, yi represents the actual values, and ŷi represents the predicted values.

The average MSE for each prediction method across body parts and feature selection methods was obtained. The average MSEs of the prediction methods were compared using the Friedman test. Pairwise comparisons were conducted with Bonferroni correction to control for multiple comparisons. Statistical significance was set at *p* < 0.05, and all analyses were performed using SPSS software (IBM, New York, NY, USA; version 29.0.02).

## 3. Results

Table 1 shows an overview of the features selected utilizing six different feature selection algorithms for the temperature of four body parts. As shown in Table 2, temperatures recorded pre-bath, 20 min post-bath, and 30 min post-bath were selected more frequently (over four times) than other features across various algorithms and temperature categories. Additionally, 10 min post-bath temperature measurements were particularly influential on left-hand deep temperature, whereas bath temperature and bathing time were significant predictors of right-hand skin temperature.

Table 3 summarizes the MSE for each prediction model and feature selection method. ANFIS exhibited the highest predictive accuracy for right-hand skin temperature, achieving the lowest MSE (0.028 °C) when combined with the RF + RFE feature selection method, while EVOFIS (0.033 °C) and RF (0.035 °C) also performed well but were slightly less effective. For left-hand deep temperature, EVOFIS paired with the L2 feature selection method achieved the lowest MSE (0.038 °C). In predicting left-hand non-contact surface temperature, ANFIS performed relatively well, with an MSE of 0.091 °C when using the L2 feature selection method, though its accuracy was lower than that observed for other body parts. In forehead skin temperature prediction, SVM outperformed all other methods, achieving the lowest MSE (0.047 °C) with the L2 feature selection method, while EVOFIS also delivered competitive results (0.049 °C).

Figure 3 shows that the models with the lowest MSEs for each body part (bold values in Table 3) exhibit a strong linear correlation between predicted and actual values, as evidenced by high R^2^ values of 0.92 for right-hand skin temperature (Figure 3a), 0.85 for left-hand deep temperature (Figure 3b), 0.67 for left-hand non-contact surface temperature (Figure 3c), and 0.90 for forehead skin temperature (Figure 3d). The *p*-values are all less than 0.01. Consistent with the MSE results, the R^2^ value for left-hand non-contact surface temperature is the lowest, indicating a relatively weak correlation compared to other temperature categories.

Figure 4 presents a visual comparison between the predicted and actual 60 min post-bath body temperatures across the four body regions. The predictions were generated using the best-performing model for each region, as identified in the previous sections. Overall, the predicted values exhibit strong agreement with the actual measurements, with particularly high accuracy observed in the right-hand skin and forehead skin temperature predictions. The prediction for left-hand deep temperature also demonstrates reasonable consistency, although some fluctuations are evident. In contrast, the left-hand non-contact surface temperature displays relatively large deviations. These visual findings are consistent with the quantitative evaluation results reported in Table 3 and Figure 3.

Figure 5 illustrates the membership function graphs for the best-performing models (bold values in Table 3) and feature selection methods. For ANFIS (Figure 5a,c), the membership function graphs focus on partitioning individual input variables. Each graph represents a specific feature, effectively segmenting the input space into distinct regions. In contrast, EVOFIS (Figure 5b) employs a holistic approach, where its membership functions integrate all input variables within specific fuzzy rules. Each graph corresponds to a single fuzzy rule that encapsulates the complex interactions among all features.

The Gaussian-shaped membership functions observed for variables such as bath temperature (e.g., Figure 5a, 32–36 °C) indicate that the model identifies specific value ranges as particularly informative for prediction. These fuzzy regions serve as the basis for rule evaluation and output inference in the system.

Figure 6 illustrates the convergence curves for the best-performing models. Both ANFIS (Figure 6a,c) and EVOFIS (Figure 6b) models effectively reduce their loss values and achieve stable convergence by the end of training. While ANFIS exhibits relatively fast convergence in the early stages of training, EVOFIS demonstrates a more gradual but steady reduction in loss values.

Figure 7 presents the statistical analysis of MSE. EVOFIS demonstrated a significantly lower average MSE than ANFIS (*p* = 0.001), eTS (*p* < 0.001), and RF (*p* < 0.001). SVM showed a significantly lower MSE than eTS (*p* = 0.001) and RF (*p* = 0.002), with no significant difference from EVOFIS or ANFIS.

## 4. Discussion

This study focuses on the prediction of post-bath body temperature using fuzzy inference systems. Advanced FIS models (ANFIS, EVOFIS, and eTS) were utilized in this research to capture the complex, nonlinear, and time-dependent dynamics of temperature changes influenced by features such as bath temperature and duration. Furthermore, these models were compared to traditional machine learning approaches, including RF and SVM, highlighting the superior accuracy and adaptability of FIS-based methods, particularly EVOFIS.

Feature selection analysis revealed that the most frequently selected predictors varied depending on the body part being examined. However, temperatures recorded pre-bath, 20 min post-bath, and 30 min post-bath were consistently identified as critical features for all body parts, indicating their significant influence on final temperature outcomes. This finding is expected, as the pre-bath temperature serves as a baseline, while 20- and 30 min post-bath temperatures provide the most relevant time-point data for predicting the target temperature at 60 min. Additionally, 10 min post-bath temperature measurements were selected for left-hand deep temperature more frequently than others, while bath temperature, and bathing time were influential on right-hand skin temperature. These results emphasize the complexity of feature interactions in temperature prediction.

From an overall perspective, while the SVM model shows comparable performance to ANFIS, the EVOFIS model exhibits the best performance, achieving the lowest average MSE values. This finding suggests that the FIS is well-suited for modeling the nonlinear and intricate relationships associated with hydrotherapy-induced body temperature changes. Notably, SVM outperformed RF, likely due to its kernel-based approach, which partially addresses nonlinearity but lacks the interpretability (e.g., membership functions or rule charts) of fuzzy logic systems.

The membership functions derived from ANFIS and EVOFIS provide insight into how each model interprets thermal and temporal parameters associated with human thermoregulation. EVOFIS, which integrates multiple input variables into each fuzzy rule, is capable of capturing complex interactions. In contrast, ANFIS employs variable-specific membership functions with well-defined thresholds for individual features. These observations suggest that the fuzzy representations learned by both models not only contribute to predictive accuracy but also provide interpretable physiological insights.

EVOFIS demonstrated the best predictive accuracy for the body temperatures of different body parts, particularly for left-hand deep temperature, achieving the lowest MSE (0.038 °C) among all models and methods. This can be attributed to EVOFIS’s ability to dynamically evolve its rule base and fine-tune its membership functions, making it well suited for capturing the complex and nonlinear relationships inherent in internal physiological processes. In contrast, ANFIS outperformed other models in predicting surface temperature, such as right-hand skin temperature (MSE = 0.028 °C), possibly due to its predefined structure and efficient training mechanism, which is effective for modeling more stable and directly observable surface responses. In contrast, SVM showed relatively strong performance in predicting forehead skin temperature, which may be due to the relatively linear and stable thermal response of the forehead surface. Compared to deep body temperature, forehead temperature might follow more direct and less variable patterns, making it more amenable to kernel-based modeling. This result suggests that SVM may be better suited for simpler surface temperature dynamics, though further investigation would be needed to confirm this. These findings may suggest that surface temperature prediction tends to benefit from simpler and more structured models such as ANFIS, while deep body temperature prediction might require more adaptive and flexible approaches such as EVOFIS, which could better account for temporal delays and complex feature interactions. However, further validation with larger and more diverse datasets would be necessary to confirm these trends. Moreover, non-contact surface temperature predictions for the left hand showed higher errors across all models. This reduced accuracy may be due to the inherent susceptibility of non-contact measurements to sensor noise, environmental variation, and inconsistencies in measurement angle or distance, all of which can introduce significant variability and reduce model reliability [23]. Factors such as differences in the distance between the sensor and the skin surface may significantly influence measurement accuracy [24]. To address these challenges, future research should focus on optimizing the precision and consistency of non-contact measurement techniques, thereby further enhancing the predictive performance of the models.

Compared with previous research, the present study provides several novel contributions. Prior studies on body temperature prediction have predominantly focused on the real-time estimation of current thermal states, typically using surface temperature measurements and conventional regression models. For instance, Limpabandhu et al. [8] employed infrared thermography for facial temperature estimation, achieving an error margin of ±0.285 °C, while Niedermann et al. [9] used non-contact skin and heat flux data with similarly high accuracy. However, these approaches have been largely limited to instantaneous or short-term predictions and did not consider post-immersion temperature dynamics, which are particularly important in hydrotherapy settings. In contrast, our study explored future temperature prediction up to 60 min after bathing using FISs—a comparatively underutilized methodology in this domain. Our results demonstrate the potential of FIS models, particularly EVOFIS, to effectively capture the complex, nonlinear, and delayed thermal responses associated with deep body temperature regulation. These findings not only complement existing research but also address critical gaps in predictive modeling for hydrotherapy safety, offering a more interpretable and adaptive alternative to conventional AI techniques.

This study has several limitations. First, earlier points of post-bath temperatures (e.g., 10, and 20 min) were used for the prediction of post-bath body temperature due to the limited features of the records in this study. The model should integrate only pre-bath features, including intrinsic features (e.g., blood test results) and extrinsic features (e.g., weather conditions, bathing duration, type of bath additives, or hot spring components) for practical applications. Also, not only 60 min temperature prediction showed in our study, but time series temperature prediction with only pre-bath should also be developed in future work. Second, further research should optimize the impact of the number of selected features. Third, hyperparameter settings and training data selection may have influenced the results, necessitating further fine-tuning and cross-validation. Additionally, the relatively small training dataset (180 records) raises concerns about potential overfitting. Furthermore, the dataset was collected between 1991 and 1997, which may limit the temporal relevance and generalizability of the findings to current clinical contexts. Expanding the dataset to include more diverse populations and varied environmental conditions will be critical for improving model generalizability and scalability.

This study demonstrated the effectiveness of FIS-based methods, particularly EVOFIS, in predicting post-bath deep body temperature. FIS models were compared to traditional machine learning approaches, and the results highlight the superior accuracy and adaptability of FISs in capturing nonlinear physiological dynamics. Additionally, these findings suggest the broader applicability of FIS-based models in predicting biological variables in hydrotherapy, where complex and nonlinear interactions are common. Beyond model performance, the ability to accurately predict post-bath body temperature—especially in deep tissue—could support the early detection of hyperthermia risk, enhancing safety for vulnerable populations such as the elderly and individuals with pre-existing conditions. Furthermore, the interpretability of FIS outputs may assist healthcare practitioners in customizing bathing protocols, such as adjusting water temperature or immersion time, based on individual physiological profiles. This approach could also be extended to home-based or remote hydrotherapy settings, enabling safer and more personalized care through data-driven decision-support tools.

## 5. Conclusions

In this study, FIS methodologies were employed to predict post-bath body temperatures, both surface and deep, and the predictive performance of various FIS models was compared to that of traditional machine learning approaches. The findings indicate that FISs, particularly EVOFIS, are highly effective in predicting post-bath body temperature, addressing the critical need for accurate and non-invasive monitoring in hydrotherapy. The high interpretability of FISs, in contrast to traditional machine learning methods, facilitated the identification of key predictors influencing body temperature dynamics. These results highlight the potential of FISs as a potential tool for body temperature prediction in hydrotherapy. Future studies should focus on integrating additional physiological and environmental features to enhance model accuracy and applicability.

## Figures and Tables

**Figure 1 healthcare-13-00972-f001:**
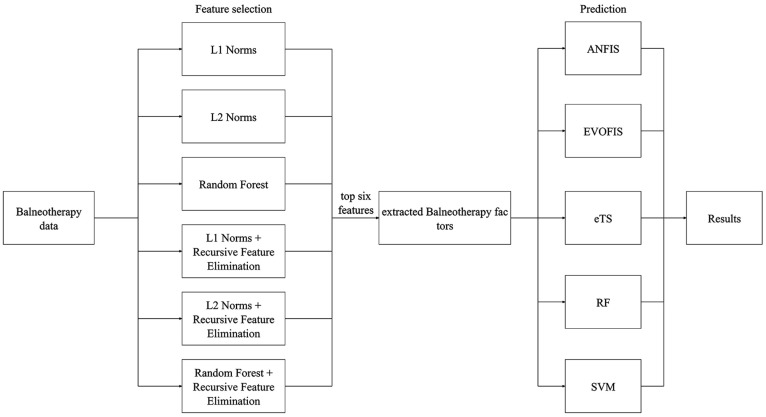
Analysis flowchart.

**Figure 2 healthcare-13-00972-f002:**
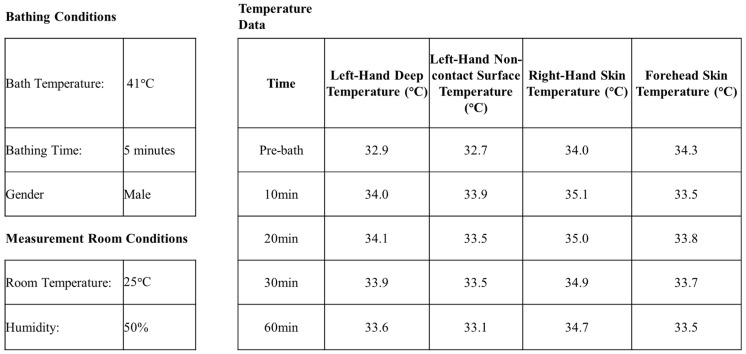
Data from a subject.

**Figure 3 healthcare-13-00972-f003:**
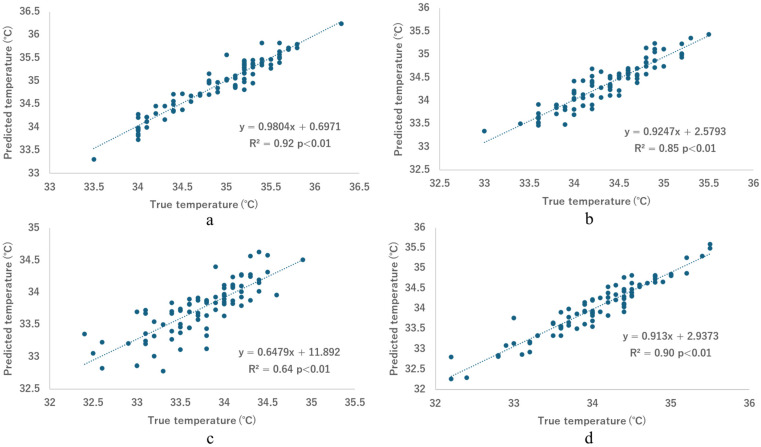
Predicted temperature versus true temperature. (**a**) Right-hand skin temperature, predicted using ANFIS with random forest and recursive feature elimination; (**b**) left-hand deep temperature, predicted using EVOFIS with L2-based feature selection; (**c**) left-hand non-contact surface temperature, predicted using ANFIS with L2-based feature selection; and (**d**) forehead skin temperature, predicted using SVM with L2-based feature selection.

**Figure 4 healthcare-13-00972-f004:**
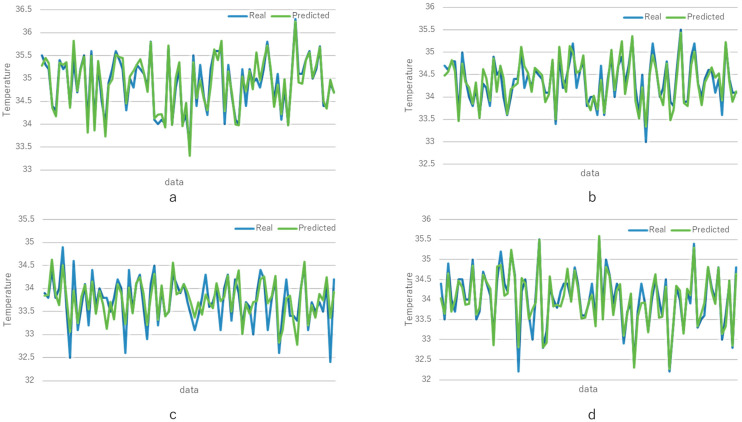
Comparison of real and predicted post-bath temperatures over test samples. (**a**) Right-hand skin temperature, predicted using ANFIS with random forest and recursive feature elimination; (**b**) left-hand deep temperature, predicted using EVOFIS with L2-based feature selection; (**c**) left-hand non-contact surface temperature, predicted using ANFIS with L2-based feature selection; and (**d**) forehead skin temperature, predicted using SVM with L2-based feature selection.

**Figure 5 healthcare-13-00972-f005:**
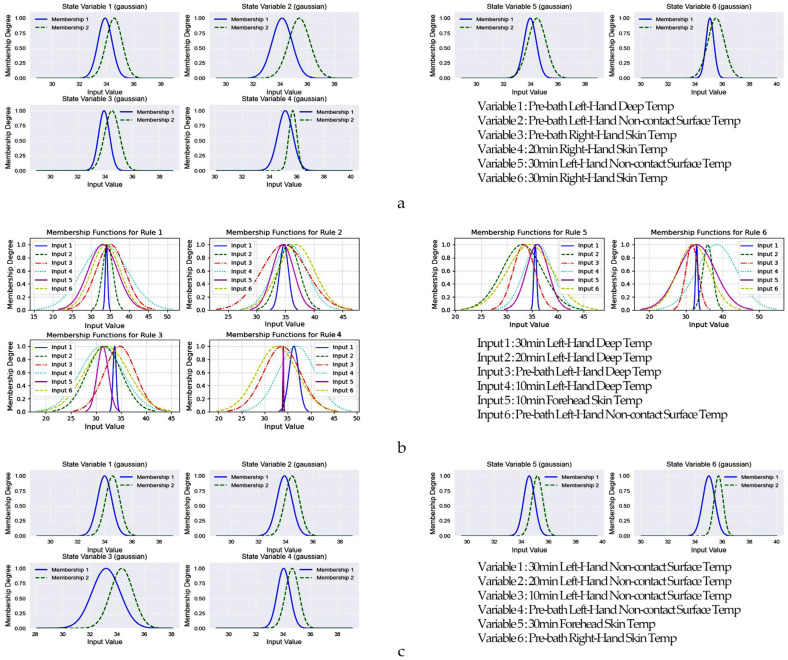
Membership function graphs for FIS models. (**a**) Right-hand skin temperature (RF + RFE + ANFIS); (**b**) left-hand deep temperature (L2 + EVOFIS); (**c**) left-hand non-contact surface temperature (L2 + ANFIS).

**Figure 6 healthcare-13-00972-f006:**
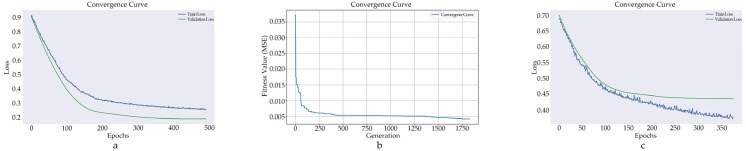
Convergence curves of FIS models. The left (**a**) and right (**c**) plots show the training and validation loss curves for models trained using ANFIS. The middle (**b**) represents the fitness value or error reduction over generations for the EVOFIS model.

**Figure 7 healthcare-13-00972-f007:**
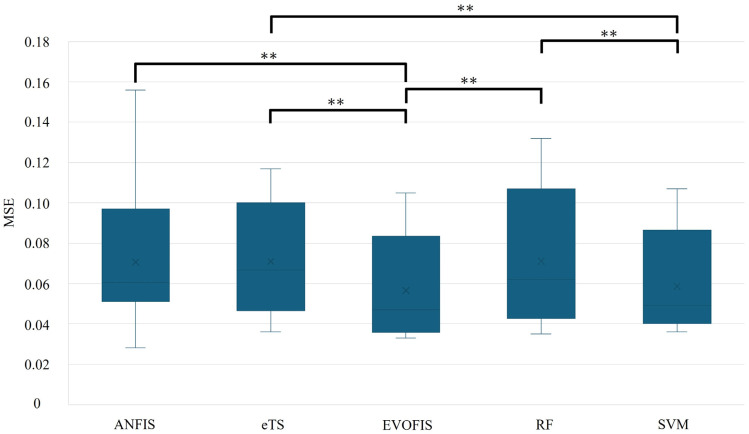
Box plot of MSE comparison across prediction models (**, *p* < 0.01), where ‘x’ represents the average value.

**Table 1 healthcare-13-00972-t001:** Feature selection results.

Target Variable	60 min Right-Hand Skin Temperature						60 min Left-Hand Deep Temperature					
Feature Selection Algorithm	L1	L1 + RFE	L2	L2 + RFE	RF	RF + RFE	L1	L1 + RFE	L2	L2 + RFE	RF	RF + RFE
Selected features	30 min right-hand skin temp	30 min right-Hand skin temp	30 min right-hand skin temp	Pre-bath left-Hand deep temp	Pre-bath left-hand deep temp	Pre-bath left-Hand deep temp	30 min left-hand deep temp	30 min left-hand deep temp	30 min left-hand deep temp	Pre-bath left-Hand deep temp	Pre-bath left-hand deep temp	Pre-bath left-hand deep temp
	Pre-bath forehead skin temp	30 min forehead skin temp	20 min right-hand skin temp	Pre-bath right-hand skin temp	Pre-bath forehead skin temp	Pre-bath left-Hand non-contact surface temp	Pre-bath left-hand deep temp	20 min left-hand deep temp	20 min left-hand deep temp	Pre-bath right-hand skin temp	Pre-bath forehead skin temp	10 min left-hand deep temp
	Bath temp	Pre-bath forehead skin temp	Pre-bath right-hand skin temp	Pre-bath forehead skin temp	20 min forehead skin temp	Pre-bath right-hand skin temp	20 min left-hand deep temp	Pre-bath left-Hand deep temp	Pre-bath left-hand deep temp	Pre-bath forehead skin temp	10 min left-hand deep temp	10 min forehead skin temp
	Pre-bath right-hand skin temp	Pre-bath left-hand deep temp	Pre-bath left-hand deep temp	30 min right-hand skin temp	30 min left-hand deep temp	20 min right-hand skin temp	Pre-bath forehead skin temp	Pre-bath forehead skin temp	10 min left-hand deep temp	20 min left-hand deep temp	20 min left-hand deep temp	20 min left-hand deep temp
	Pre-bath left-hand deep temp	20 min forehead skin temp	30 min left-hand non-contact surface temp	Bath temp	30 min right-hand skin temp	30 min left-hand non-contact surface temp	10 min left-hand non-contact surface temp	10 min left-hand deep temp	10 min forehead skin temp	30 min left-hand deep temp	30 min left-hand deep temp	20 min left-hand non-contact surface temp
	Bathing time	30 min left-hand deep temp	Pre-bath left-hand non-contact surface temp	Bathing time	30 min forehead skin temp	30 min right-hand skin temp	30 min forehead skin temp	30 min forehead skin temp	Pre-bath left-hand non-contact surface temp	30 min forehead skin temp	30 min forehead skin temp	30 min left-hand deep temp
**Target** **Variable**	**60 min Left-Hand Non-Contact Surface Temperature**						**60 min Forehead Skin Temperature**					
**Feature** **Selection Algorithm**	**L1**	**L1 + RFE**	**L2**	**L2 + RFE**	**RF**	**RF + RFE**	**L1**	**L1 + RFE**	**L2**	**L2 + RFE**	**RF**	**RF + RFE**
Selected features	30 min left-hand non-contact surface temp	30 min left-hand non-contact surface temp	30 min left-hand non-contact surface temp	Pre-bath left-hand non-contact surface temp	Pre-bath left-hand non-contact surface temp	Pre-bath left-hand non-contact surface temp	30 min forehead skin temp	30 min forehead skin temp	30 min forehead skin temp	Pre-bath left-hand deep temp	Pre-bath left-hand deep temp	Pre-bath left-hand deep temp
	20 min left-hand non-contact surface temp	20 min left-hand non-contact surface temp	20 min left-hand non-contact surface temp	Pre-bath right-hand skin temp	10 min left-hand non-contact surface temp	Pre-bath right-hand skin temp	20 min forehead skin temp	20 min forehead skin temp	20 min forehead skin temp	Pre-bath left-hand non-contact surface temp	Pre-bath left-hand non-contact surface temp	Pre-bath forehead skin temp
	Pre-bath left-hand non-contact surface temp	Pre-bath left-hand non-contact surface temp	10 min left-hand non-contact surface temp	20 min left-hand non-contact surface temp	20 min left-hand non-contact surface temp	10 min left-hand non-contact surface temp	Pre-bath left-hand non-contact surface temp	30 min left-hand non-contact surface temp	10 min forehead skin temp	20 min forehead skin temp	20 min left-hand non-contact surface temp	10 min forehead skin temp
	30 min left-hand deep temp	10 min left-hand non-contact surface temp	Pre-bath left-hand non-contact surface temp	30 min left-hand deep temp	30 min left-hand deep temp	20 min left-hand non-contact Surface temp	30 min left-hand deep temp	Pre-bath left-hand non-contact surface temp	Pre-bath forehead skin temp	30 min left-hand deep temp	20 min forehead skin temp	20 min left-hand deep temp
	30 min right-hand skin temp	30 min right-hand skin temp	30 min forehead skin temp	30 min left-hand non-contact surface temp	30 min left-hand non-contact surface temp	30 min left-hand non-contact surface temp	Pre-bath left-hand deep temp	20 min left-hand non-contact surface temp	Pre-bath left-hand deep temp	30 min forehead skin temp	30 min left-hand non-contact surface temp	20 min forehead skin temp
	Pre-bath right-hand skin temp	10 min forehead skin temp	Pre-bath right-hand skin temp	30 min right-hand skin temp	30 min right-hand skin temp	30 min forehead skin temp	Bathing time	20 min right-hand skin temp	Pre-bath left-hand non-contact surface temp	Bathing time	30 min forehead skin temp	30 min forehead skin temp

L1: L1-based selection; L1 + RFE L1 combined with recursive feature elimination (RFE); L2: L2-based selection; L2 + RFE: L2 combined with RFE; RF: random forest-based selection; RF + RFE: random forest (RF) combined with RFE.

**Table 2 healthcare-13-00972-t002:** Feature selection counts.

60 min Right-Hand Skin Temperature		60 min Left-Hand Deep Temperature		60 min Left-Hand Non-Contact Surface Temperature		60 min Forehead Skin Temperature	
Feature	Counts	Feature	Counts	Feature	Counts	Feature	Counts
30 min right-hand skin temp	6	30 min left-hand deep temp	6	30 min left-hand non-contact surface temp	6	30 min forehead skin temp	6
Pre-bath left-hand deep temp	6	Pre-bath left-hand deep temp	6	Pre-bath left-hand non-contact surface temp	6	20 min forehead skin temp	6
Pre-bath forehead skin temp	4	20 min left-hand deep temp	6	20 min left-hand non-contact surface temp	5	Pre-bath left-hand non-contact surface temp	5
Pre-bath right-hand skin temp	4	Pre-bath forehead Skin Temp	4	30 min right-hand skin temp	4	Pre-bath left-hand deep temp	5
Bath temp	2	30 min forehead skin temp	4	Pre-bath right-hand skin temp	4	Bathing time	2
Bathing time	2	10 min left-hand deep temp	4	10 min left-hand non-contact surface temp	4	30 min left-hand non-contact surface temp	2
30 min forehead skin temp	2	10 min forehead skin temp	2	30 min forehead skin temp	2	20 min left-hand non-contact surface temp	2
20 min forehead skin temp	2	10 min left-hand non-contact surface temp	1	30 min left-hand deep temp	2	10 min forehead skin temp	2
30 min left-hand deep temp	2	Pre-bath left-hand non-contact surface temp	1	20 min left-hand non-contact surface temp	1	Pre-bath forehead skin temp	2
20 min right-hand skin temp	2	Pre-bath right-hand skin temp	1	30 min left-hand deep temp	1	30 min left-hand deep temp	1
30 min left-hand non-contact surface temp	2	20 min left-hand non-contact surface temp	1	10 min forehead skin temp	1	20 min right-hand skin temp	1
Pre-bath left-hand non-contact surface temp	2					30 min left-hand deep temp	1
						20 min left-hand deep temp	1

**Table 3 healthcare-13-00972-t003:** Results of mean squared error.

	Right-Hand Skin Temperature (°C)							Left-Hand Deep Temperature (°C)						
	L1	L1 + RFE	L2	L2 + RFE	RF	RF + RFE	Average	L1	L1 + RFE	L2	L2 + RFE	RF	RF + RFE	Average
ANFIS	0.040	0.036	0.040	0.061	0.035	**0.028**	0.040	0.058	0.051	0.070	0.057	0.069	0.051	0.059
eTS	0.041	0.040	0.046	0.036	0.036	0.036	0.039	0.070	0.082	0.074	0.048	0.082	0.082	0.073
EVOFIS	0.034	0.033	0.034	0.035	0.033	0.035	0.034	0.045	0.044	**0.038**	0.044	0.044	0.045	0.043
RF	0.036	0.036	0.042	0.040	0.035	0.035	0.037	0.045	0.047	0.055	0.057	0.059	0.060	0.053
SVM	0.037	0.036	0.038	0.039	0.037	0.037	0.037	0.043	0.043	0.048	0.044	0.049	0.049	0.046
	**Left-Hand Non-Contact Surface Temperature** **(°C)**							**Forehead Skin Temperature** **(°C)**						
	**L1**	**L1 + RFE**	**L2**	**L2 + RFE**	**RF**	**RF + RFE**	**Average**	**L1**	**L1 + RFE**	**L2**	**L2 + RFE**	**RF**	**RF + RFE**	**Average**
ANFIS	0.099	0.101	**0.091**	0.099	0.111	0.106	0.101	0.060	0.056	0.069	0.060	0.091	0.156	0.082
eTS	0.109	0.106	0.109	0.109	0.117	0.117	0.111	0.063	0.063	0.053	0.052	0.069	0.064	0.061
EVOFIS	0.102	0.099	0.094	0.096	0.105	0.098	0.099	0.049	0.050	0.050	0.052	0.049	0.049	0.050
RF	0.120	0.119	0.118	0.119	0.132	0.132	0.123	0.073	0.074	0.070	0.068	0.064	0.069	0.070
SVM	0.102	0.107	0.102	0.102	0.098	0.098	0.101	0.052	0.052	**0.047**	0.049	0.049	0.049	0.050

L1: L1-based selection; L1 + RFE L1 combined with recursive feature elimination (RFE); L2: L2-based selection; L1 + RFE: L2 combined with RFE; RF: random forest-based selection; RF + RFE: and random forest (RF) combined with RFE. ANFIS: adaptive neuro-fuzzy inference system; eTS: evolving Takagi-Sugeno system; EVOFIS: evolving fuzzy inference system; SVM: support vector machine. The bold values indicate the lowest MSE for each measured body part.

## Data Availability

The data that support the findings of this study are available from the corresponding author upon reasonable request.

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
