# Peer review of "Prediction of Post-Bath Body Temperature Using Fuzzy Inference Systems with Hydrotherapy Data"

_healthcare, 2025, doi:10.3390/healthcare13090972_

Round 1
Reviewer 1 Report
Comments and Suggestions for Authors
REVIWER’s Comments
Manuscript Number: healthcare-3563874
Title: Body Temperature Prediction of Post-bath Using Fuzzy Infer-2 ence Systems with Hydrotherapy Data
Reviewer Comments
This study examines Fuzzy Inference Systems (FIS) for predicting post-bath body temperature in hydrotherapy, ensuring safety and therapeutic optimization. Using models like EVOFIS, ANFIS, and Enhanced Takagi-Sugeno, comparisons with Random Forest and SVM revealed EVOFIS as the most effective, particularly for deep body temperature prediction. The findings highlight the potential of FIS-based models for non-invasive monitoring, benefiting vulnerable populations. This research advances personalized hydrotherapy safety and efficiency. Major Comments
- The text in Figure 1 is barely visible and must be enhanced for better readability.
- The dataset used is outdated. The authors should clarify the reason for selecting data from these specific years.
- The data in Table 1 is difficult to understand. It should be restructured for better readability.
- The authors need to provide a sample dataset along with its attribute ranges:
The number of rows and columns in the dataset should be mentioned.
b. A description of each column should be provided. - Critical values are missing in each experiment and should be included.
- The purpose of displaying empty graphs in Figure 4(a) and 4(c) (right side) is unclear. The authors should justify their inclusion.
- The text in Figure 5 is barely visible and should be redrawn for better readability.
- The ML system architecture is missing and should be included.
- The authors have not discussed the impact of bathing temperatures after 60 minutes.
- How does it significantly affect blood flow and other physiological aspects?
- Why was 60 minutes chosen as the target value?
- There is no clear discussion on the FIS methodologies used to predict post-bath body temperature at the surface and deep levels.
- The section explaining deep temperature changes is unclear and should be explicitly mentioned.
- The authors have not provided statistical evidence to validate the results.
- A benchmarking table comparing the proposed approach with existing methods is missing.
- The paper lacks a discussion on the literature review and existing related works.
- The final results are not well visualized and should be presented in a more readable and understandable manner.
Comments on the Quality of English Language
The manuscript should be proofread by a professional English language expert or a reputable proofreading agency to ensure accuracy and clarity
Reviewer 2 Report
Comments and Suggestions for Authors
The manuscript presents a novel application of Fuzzy Inference Systems (FIS) for predicting post-bath body temperature, comparing FIS-based models (ANFIS, EVOFIS, eTS) with traditional machine learning methods (RF, SVM). While the study is well-structured and addresses an important clinical need, several weaknesses require attention to enhance clarity, methodological rigor, and practical applicability.
Below are my concerns:
Title and Abstract (Page 1)
The abstract could better highlight the clinical implications of the findings. For example, emphasize the clinical significance of EVOFIS’s superiority in the abstract.
Moreover, keywords are standard but could include "non-invasive monitoring" or "physiological prediction" for broader reach. Expand keywords to improve discoverability.
Introduction (Pages 1–2)
It has a limited discussion on why existing methods (e.g., infrared thermography) fall short for predictive (vs. real-time) tasks. Moreover, the rationale for focusing on 60-minute post-bath temperature is not explicitly justified.
Please, clarify why predicting 60-minute temperature is clinically critical (e.g., hyperthermia risks). Also, contrast FIS’s interpretability with "black-box" AI methods more explicitly.
Materials and Methods (Pages 3–5)
Data Limitations: I am worried about your generalizability because you are considering a small dataset (213 records) and outdated (1991–1997). Also, address dataset limitations (e.g., potential biases, temporal relevance).
Feature Selection: No justification for using six different methods or how inconsistencies were resolved. Explain how feature selection discrepancies were handled (e.g., voting system?).
Model Training: Lack of discussion on cross-validation strategies to mitigate overfitting. Include k-fold cross-validation results to support robustness.
Results (Pages 5–9)
Membership functions (Figure 4) are described but not linked to physiological insights.
In addition, SVM outperforms for forehead temperature; this discrepancy is not discussed. Please discuss why SVM excels for forehead temperature (e.g., linearity of skin temperature dynamics?).
Furthermore, Poor performance for left-hand non-contact temperature is noted but not explored. Analyze why non-contact measurements underperform (e.g., sensor noise, anatomical variability).
Discussion (Pages 11–12)
Overstates FIS’s interpretability without showing actionable rules or clinical thresholds. Provide examples of fuzzy rules or thresholds for clinical decision-making.
Does not compare results with prior works (e.g., MSE of ±0.49°C in [10] vs. 0.038°C here). Benchmark against existing prediction errors in the literature.
Conclusions (Page 12–13)
Lacks specific recommendations for clinical implementation. Propose a roadmap for integrating FIS into hydrotherapy protocols.
Future work is generic (e.g., "expand dataset"). Suggest collaborations to collect larger, diverse datasets.
Reviewer 3 Report
Comments and Suggestions for Authors
- To ensure reproducibility of the experiment, the manuscript should explicitly outline the parameter settings, software tools, and statistical methods used in the study to ensure reproducibility by other researchers. Consider providing a supplementary document or appendix with detailed implementation steps would be beneficial.
- Why the current approach is selected (FIS method), a clearer justification is needed on why these methods were chosen over alternative approaches. A comparative analysis or citation of prior studies that used similar methodologies would strengthen the argument.
- Since the dataset is collected retrospectively, consider providing details on the data sources, including how the datasets were curated, their inclusion/exclusion criteria, and any preprocessing steps taken to ensure data consistency and reliability.
- Is ethical clearance obtained from the relevant department/ institution? Since the dataset is obtained retrospectively.
- Some findings, especially on the implications in a real-world healthcare setting lacks in-depth discussion. Consider expanding the current discussion on how the results translate into clinical applications and potential benefits for practitioners.
- Consider including limitations and future works in the manuscript.
Round 2
Reviewer 1 Report
Comments and Suggestions for Authors
The authors have addressed all the concerns and suggestions raised in the previous review thoroughly. The manuscript has been significantly improved in terms of clarity, data presentation, and methodological rigor. Specific issues such as figure readability, dataset explanation, missing experimental details, and the lack of system architecture have been satisfactorily resolved.
A comprehensive literature review has been added, and statistical validation has been incorporated to support the results. The inclusion of benchmarking comparisons further strengthens the quality of the work.
Given the substantial improvements and the completeness of the revised manuscript, I recommend acceptance of the paper in its current form.
Reviewer 2 Report
Comments and Suggestions for Authors
The authors have addressed every point raised in the previous review. I have no further comments and suggest approval.